# Small and Random Peptides: An Unexplored Reservoir of Potentially Functional Primitive Organocatalysts. The Case of Seryl-Histidine

**DOI:** 10.3390/life7020019

**Published:** 2017-04-09

**Authors:** Rafal Wieczorek, Katarzyna Adamala, Tecla Gasperi, Fabio Polticelli, Pasquale Stano

**Affiliations:** 1Faculty of Chemistry, University of Warsaw, Pasteura 1, 02-093 Warsaw, Poland; wieczorek@chem.uw.edu.pl; 2Department of Genetics, Cell Biology, and Development, University of Minnesota, Minneapolis, MN 55455, USA; kadamala@umn.edu; 3Department of Science, Roma Tre University, Viale G. Marconi 446, 00146 Rome, Italy; tecla.gasperi@uniroma3.it (T.G.); polticel@uniroma3.it (F.P.); 4National Institute of Nuclear Physics, Roma Tre Section, Via della Vasca Navale 84, 00146 Rome, Italy; 5Department of Biological and Environmental Sciences and Technologies (DiSTeBA), University of Salento, Campus Ecotekne (S.P. 6 Lecce-Monteroni), 73100 Lecce, Italy

**Keywords:** fragment condensation, organocatalysis, peptide bond formation, phosphodiester bond formation, Ser-His, small peptides

## Abstract

Catalysis is an essential feature of living systems biochemistry, and probably, it played a key role in primordial times, helping to produce more complex molecules from simple ones. However, enzymes, the biocatalysts par excellence, were not available in such an ancient context, and so, instead, small molecule catalysis (organocatalysis) may have occurred. The best candidates for the role of primitive organocatalysts are amino acids and short random peptides, which are believed to have been available in an early period on Earth. In this review, we discuss the occurrence of primordial organocatalysts in the form of peptides, in particular commenting on reports about seryl-histidine dipeptide, which have recently been investigated. Starting from this specific case, we also mention a peptide fragment condensation scenario, as well as other potential roles of peptides in primordial times. The review actually aims to stimulate further investigation on an unexplored field of research, namely one that specifically looks at the catalytic activity of small random peptides with respect to reactions relevant to prebiotic chemistry and early chemical evolution.

## 1. Introduction

In addition to inorganic factors, such as mineral surfaces [1,2], light [3], the reducing environment of hydrothermal vents [4] and spark discharges [5], the role of organic molecules as early catalysts has been widely recognized in origin of life scenarios. Catalysis is omnipresent in modern life, and it could be argued that catalysis was one of the key mechanisms contributing to the origins of life. However, it is not easy to determine which primitive catalysts were actually present and what their role would have been; nor is it any easier to determine in which phase of prebiotic evolution they would have emerged or how they could have contributed to the development from the primordial chemical world to the birth of the first living cells.

In this respect, the roles of small organic molecules, which arose in abiotic syntheses [6], RNA [7] or other simple nucleic acids [8], and small peptides [9,10,11], whose chemical formation is viable under various routes [12], have been discussed. The catalytic role of RNA in the modern ribosome has been adopted to validate the theory of a pure ‘RNA World’ [13], but the well-known chemical instability of RNA suggests more complex scenarios of the co-evolution of RNA and small peptides [11,14]. For example, recent theoretical investigations based on the activity and function of aminoacyl-tRNA suggest that an RNA-only metabolism may never have existed [15].

Since the 1953 seminal work by Stanley Miller on the abiogenesis of amino acids [16], prebiotic chemistry [17] focuses on revealing how some organics of biological relevance can be formed via plausible chemical routes from simple primitive molecules. Once formed, some of these available molecules could have acted: (i) as catalysts and/or building blocks in early metabolic cycles; and (ii) as catalysts for the synthesis of large chain-like molecules, i.e., proteins and nucleic acids. For the origin of life, both the onset of reaction networks and the transition from small, low molecular weight compounds to macromolecules are decisive.

Since we previously focused our attention on the role of peptides in these prebiotic scenarios [18,19,20], here we aim to discuss the literature regarding the discovery that a simple dipeptide, Ser-His (Figure 1), is capable of catalyzing the coupling reaction of amino acid derivatives or nucleotide derivatives. We will then present a general view on what could have been the roles of short and random peptides in the origins of life. In doing this, we will cite selected papers of the origins of life literature. It should be clear that this review is not intended to be comprehensive with respect to the large amount of work done on this subject. Some references will be just briefly commented on, whereas only the most relevant for our discussion will be presented in detail. In addition to Ser-His-related findings, screening of peptide libraries, organocatalysis mechanisms, molecular recognition and interaction with primitive membranes are possible fields of inquiry for further development in this research.

The paper is organized into five sections. Firstly, we introduce organocatalysis, as it is known today in organic chemistry (Section 2) since modern chemical approaches that are often used to synthesize complex molecules with high chemo-, regio- and stereo-selectivity rely on the same mechanisms that probably contributed to the emergence of biomolecules and cellular life billions years ago. After a quick review of the current ideas about the prebiotic origin of amino acids and peptides (Section 3), we will follow with a detailed discussion of the two important synthetic reactions catalyzed by Ser-His, namely the formation of peptide bonds and phosphodiester bonds (Section 4). This part will also present a discussion on the computer-generated three-dimensional models of Ser-His and its isomer His-Ser (not functioning as a catalyst). The fragment condensation model is then presented and discussed, by referring to the potential role of random peptides with ligation activity (Section 5). Finally, some recent accounts relating other potential roles of peptides in the origin of life are presented (Section 6).

The main finding discussed in this review stems from a fundamental principle in chemistry: microscopic reversibility. A catalyst of the reaction A − AB → A + B also catalyzes the reverse reaction A + B → A − B with the same (reversed) mechanism, but it cannot change the ΔG° of the reaction. One of YuFen Zhao’s works reports that Ser-His (and other short peptides, such as Cys-His, Ser-His-Gly, Ser-His-His, Ser-His-Asp, Ser-Gly-His-His, Ser-Gly-Gly-His-His) is capable of cleaving peptide and phosphodiester bonds [21], so we reasoned that Ser-His should also be able to catalyze the reverse reaction, the very same reverse reactions that lead to the formation of peptide and phosphodiester bonds, which potentially lead to long peptides and long nucleic acids. As is well known, peptide synthesis has been indeed accomplished in the laboratory by proteases with the caveat of being able to shift the thermodynamic equilibrium toward the desired products or by using activated acyl donors [22,23,24].

## 2. What is Organocatalysis? The Organic Chemist’s View

Although specific examples of organic catalysts have been known for a long time, the modern term ‘organocatalysis’ and its field of application have emerged only recently in synthetic chemistry [25] and can be defined as the use of small, low-molecular weight organic compounds, predominately composed of carbon, hydrogen, oxygen, sulfur and phosphorous, to accelerate chemical reactions. In the field of organic synthesis, the use of organocatalysis dates back to Liebig’s preparation of oxamide from dicyano and water (1860), and over the years, the number of examples highlighting the application of such a strategy has become countless. However, the real renaissance of organocatalysis in the practice of organic chemistry should be attributed to List’s breakthrough concerning the first proline-catalyzed direct asymmetric aldol reaction in 2000 (yield 97%, enantiomeric excess 96%) [26] and to the first report on an enantioselective organocatalytic Diels–Alder reaction [27].

A growing amount of publications has appeared since, and asymmetric organocatalysis has been designated the third general approach to the catalytic production of enantiomerically-pure compounds [25]. Indeed, a growing community of research groups has devoted their efforts to the development of novel organocatalytic strategies that offer a valuable effective alternative to well-established transition metal methods and enzymatic methods. In organic synthesis, the organocatalytic approach provides several advantages: (i) usually, the catalysts are much less expensive than the transition metal complexes; (ii) the organocatalysts are readily available, more stable and easier to synthesize; (iii) the reactions can be generally performed under low demanding conditions without the regular need for an inert atmosphere or dry solvents; and most important; (iv) different activation modes are possible with respect to substrates, reagents and reactions.

Within this context, one of the biggest advantages of organocatalysts relies on their ability to variously interact with different substrates according to the accurate choice of their chemical structure and properties. Both covalent and non-covalent interactions and activation of substrates are in the realm of organocatalysis, very often in a chemio-, regio- and stereo-selective manner. Moreover, dual and cooperative catalysis, which allows the catalyst to bind to more than one substrate by using different activation modes, has paved the way toward original reactions giving access to novel useful scaffolds. It is expected that a larger variety of organocatalysts and modes of action is going to be developed in the future. The common feature, however, will be the same: conditioning the course of a reaction by close interaction with the reaction partners, in agreement with a minimalistic version of the lock-and-key paradigm of enzyme action [28].

As this paper is devoted to organocatalysis in primitive chemical systems, readers interested in organocatalysts for modern chemical synthesis should refer to reviews and books focusing on that subject. Just to enumerate some examples, amino acids and short peptides have been used as catalysts for the following reactions: Robinson annulation, intermolecular aldol reaction, Mannich reaction, Michael addition, hydrocyanation of aldehydes and imines, epoxidation, acyl transfer reactions, conjugate additions of thiols and azides, cycloadditions, phosphorylation and a series of domino, as well as multicomponent reactions [29]. Finally, it should be noted that organocatalysis is not necessarily ‘biomimetic’.

## 3. Origin of Peptides

Amino acids are easily formed abiotically and were therefore generally available in primitive times, together with a plethora of other small organic compounds that were similarly synthesized by geochemical and environmental processes. The well-known Miller–Urey experiment, for example, produced a mixture of simple amino acids [16]. Amino acids have even been found in meteorites [30]. A consensus list of prebiotic amino acids includes Ala, Asp, Glu, Gly, Ile, Leu, Pro, Ser, Thr and Val [31,32,33,34], all of which follow a Strecker synthesis model in their mechanisms of formation [35,36,37,38].

For the discussion in this paper, we are most interested in the prebiotic availability of aspartate, serine and histidine. The first two of those amino acids form easily in simulated prebiotic conditions and have been reported in several works. Histidine, however, is not encountered as a straightforward product in, for example, Strecker-like synthesis. It has been noted, however, that once imidazole can be obtained in a prebiotic environment, it should be, in principle, possible to get to histidine through the Strecker synthesis [39]. Prebiotic imidazole syntheses have been reported in several conditions [40,41,42]. Based on the premise of the Strecker process through an imidazole intermediate, the prebiotic synthesis of histidine has been reported starting from erythrose and formamide [43]. Some have argued that such a process does not fulfil the criteria that would enable it to be considered prebiotic [42], and therefore, putting into question whether or not this amino acid could in fact be available in early stages of prebiotic evolution remains open for debate.

Even if of the availability of some amino acids is considered possible and it was demonstrated that some of them could also act as organocatalysts [44,45], their oligomerization to form early peptides is more difficult to predict. The endergonic standard Gibbs free energy change associated with the formation of peptide bonds in aqueous solutions from carboxylate and ammonium ions is about +10–20 kJ/mol [46], implying that the reaction is not thermodynamically favored. This consideration explains why investigations on primitive condensation mechanisms leading to polypeptides are challenging. Making peptide bonds plausibly required a combination of activation pathways in order to overcome and bypass this thermodynamic barrier. It is indeed remarkable that the formation of peptide bonds in the ribosome does not occur by direct condensation of two amino acid moieties. An ‘activated’ carboxylic group is employed, namely the aminoacyl-tRNA adduct, which is an amino acid ester, whose formation ultimately requires ATP energy. In biochemistry, the peptide bonds are formed via ester aminolysis, not by carboxylic acid/amine condensation.

However, the chemical path to join amino acids is not the only problem in the origin of peptides. Two others relevant issues are their chirality (e.g., were early peptides homochiral?) and the peptide sequence (did specific amino acid sequences emerge, and if so, according to which mechanism?).

Returning to the chemical aspects of amino acids oligomerization, four main routes are generally discussed: the thermal condensation in the dry state or in the liquid phase, the removal of water in dry/wet cycles, the salt-induced condensation in aqueous solution (catalyzed by cupric ions) and the oligomerization of activated amino acids (in the form of *N*-carboxyanhydrides, thioesters and other forms). In many cases, solid surfaces like clay and other minerals improve the yield and/or favor homochirality; conversely, not many studies have been devoted to the issue of peptide sequences. A detailed review of the principal results in the field has been recently published [17]. Here, we give just some examples of prebiotic routes to oligopeptides.

It was proposed that racemic mixtures of amino acid alkyl and thio-alkyl esters self-assembled on the water/air interface can undergo spontaneous condensation, forming randomly-sequenced peptide chains [47]. However, recent studies show that in the case of short side chain natural amino acids, the reaction does not proceed beyond the stage of di- and tripeptides [48], so it cannot be regarded as a prebiotic way of obtaining long and enantiopure peptides. The formation of peptides can be induced by high salt environment [49], and under certain conditions, a high enantioselective oligomerization was observed [50]. It has recently been suggested that it is possible to envision a prebiotically-plausible mechanism of stereoselective peptide bond formation based on glutathione synthesis [51]. Amino acids and other simple carboxylic acids can be synthesized in iron-catalyzed redox reactions [52]. The cyanine-rich reducing environment has been shown to yield arginine, an amino acid involved in many peptide interactions with nucleic acids [53]. Recently, it has also been demonstrated that peptidyl-RNA complexes could form under conditions simulating prebiotic RNA oligomerization and that such hybrid molecules could extend RNA’s stability, thus hinting at early reasons for RNA-peptide interactions, which eventually led to the translation system of modern life [54].

A more general, prebiotically-plausible peptide synthesis mechanism was proposed, involving 2-thiono-5-oxazolidones generated in reactions of amino acids with carbon disulfide [55] or *N*-carboxyanhydrides (Leuchs anhydrides) [56,57,58] from carbonyl sulfide [59]. Interestingly, liposomes can assist sequence-specific, stereoselective peptide formation on the surface of the bilayer [60,61]. A similar behavior has been recently reported by using fatty acid vesicles, which are prebiotically-plausible protocells [62].

These and many other results show that the presence of small peptides in early Earth was plausible. In such a large combinatorial peptide library, highly diverse in length, chirality and sequence, several interesting compounds might have been originated, so that reciprocal interactions could start. Theoretical accounts for chemical networks based on chain growing and chain degradation can be found in the work of Kauffman [63]. More attention should be given to demonstrating experimentally the onset of the Kauffman network by investigating the formation of medium-sized peptides, for example decapeptides, from a small set of reacting amino acids (or short peptides). The idea is that some of these primitive peptides that can be generated according to a Kauffman-like scheme could have catalytic properties. In particular, an interesting activity is peptide bond formation. Short random peptides could have been formed spontaneously by the above-mentioned mechanisms and thus helped the onset of protobiological mechanisms (like autocatalytic and cross-catalytic cycles, complex formation with metals or other non-peptide molecules, membrane functionalization and control of trans-membrane traffic, new peptide formation by chain elongation or fragment condensation, synthesis and stabilization of nucleic acids, and so on).

There is a prolific field of study involving self-replicating networks of helical peptide replicators as demonstrated in the labs of Ghadiri [64,65], Chmielewski [66,67] and Ashkenasy [68,69]. In these reports, it is shown that some specific α-helical or β-sheet peptides (i.e., with specific sequences) can undergo a self-replicative process, in a manner that is reminiscent, conceptually, of nucleic acids’ template-based replication. Although these works might be relevant in the broader origin of life field in the discussion on the emergence of mutually-catalytic networks (and the propagation of homochirality), a full discussion of this chemistry lies outside of the scope of this review.

Envisioning a primitive scenario with early peptide catalysts is challenging, because looking for a needle in a haystack for all possible sequences is not easy. However, such studies may lead to new chemistries with great explanatory value for the origins of life. In some cases, such peptide catalysts could alleviate the thermodynamic issues concerning condensation reactions (requiring a way to shift the equilibrium toward the products, via the removal of products or via activation of the substrates) or give a hint towards understanding early protocells. As will be discussed in the next section, Ser-His is probably the shortest peptide that has shown a catalytic capacity.

## 4. Seryl-Histidine Catalyzes the Formation of Peptide Bonds and Phosphodiester Bonds

### 4.1. The Discovery of Ser-His Hydrolase Activity

Seryl-histidine (Ser-His) and seryl-histidyl-aspartate (Ser-His-Asp) can hydrolyze esters, as well as proteins and nucleic acids [21] (but see also the note at the end of the text reporting opposite results). There are many specialized enzymes catalyzing those reactions. Most of them are serine hydrolases. Ser/His/Asp is the most common catalytic triad known from enzymatic studies [70]. It is to be found in many important enzymes, for example chymotrypsin, subtilisin or phospholipases A2. The principle behind its action has been independently invented in the course of evolution at least 23 times [71]. Serine hydrolases comprise approximately 1% of the genes in the human proteome [72].

The mechanism of the serine triad catalysis is one of the most understood of all enzymatic activities (Figure 2). The three groups act as a nucleophile (serine hydroxyl), base catalyst (histidine imidazole ring) and an orienting/stabilizing group (aspartate) [70]. Histidine is bound and stabilized by aspartate in such a way that it polarizes the nucleophile, thus allowing it to successfully perform nucleophilic attack on the substrate (usually the carbonyl carbon of a peptide). Carbonyl oxygen accepts an electron, and a tetrahedral intermediate stabilized by the oxyanion hole is formed. The electrons from carbonyl oxygen collapse back to the carbonyl carbon, ejecting half of the substrate. The other half of the substrate is still bound to the enzyme forming the acyl-enzyme intermediate. The acyl-enzyme intermediate is then attacked by another nucleophile from the environment. Briefly, another tetrahedral intermediate is formed, which then ejects the enzyme while releasing the product. If the new nucleophile is water, the result is hydrolysis, if it is another organic substrate the result is acyl transfer.

The structure and activity of Ser-His peptide, as well as its corresponding tripeptide Ser-His-Asp has prompted speculations about the mechanism of its catalytic properties. The most obvious postulate is that it is a minimalistic analogue of serine hydrolases [21]. The group of YuFen Zhao, who first reported the Ser-His activity, has followed with a number of studies on hydrolytic activity, analytics and modeling of this interesting dipeptide [73,74,75,76,77,78,79,80,81,82]. However, none of those studies have actually tried to elucidate in detail the mechanism by which Ser-His hydrolysis functions.

As often happens in research, Ser-His was discovered by serendipity. It is interesting to report here the words of Zhao and collaborators revealing how Ser-His activity was first noticed:
“In the studies of the interaction between *N*-phosphoamino acids with DNA, it was found that the aged solution of *N*-phosphorus serine in a saturated histidine buffer exhibited DNA cleavage activity, not the fresh one. Finally, it was clarified that the seryl-histidine dipeptide which formed in the solution was responsible for the DNA cleavage (Ma, 1996).”[80]


### 4.2. Peptide Bond Formation

From a prebiotic chemistry perspective, hydrolysis (proteolysis) is not a desired catalytic activity. Synthesis (condensation, reverse proteolysis) is preferred, as these reactions can drive the formation of even more complex molecules. However, as we have earlier mentioned, hydrolysis is thermodynamically favored in aqueous environments. Following the discussion presented by several authors [24,46,83], the condensation reaction between two charged amino acids (or peptides) in water is composed of two elementary steps, namely a proton exchange process and an aminolysis reaction, defined in Figure 3a ($\Delta G_{cond}^{\circ }=\Delta G_{exch}^{\circ }+\Delta G_{am}^{\circ }$). It follows that the condensation equilibrium constant ($K_{cond}$) can be written as the product between the proton exchange constant ($K_{exch}$) and the aminolysis constant ($K_{am}$). $K_{exch}$ is the ratio between the acidity constants of free amino acid terminal ammonium ion (10^−8^–10^−9^) and carboxylic acid (10^−2^–10^−3^).

(1)$$\begin{array}{cc} K_{exch} & =\frac{\left[\mathrm{RCOOH}\right]\left[R^{\prime }{\mathrm{NH}}_{2}\right]}{\left[{\mathrm{RCOO}}^{-}\right]\left[R^{\prime }{{\mathrm{NH}}_{3}}^{+}\right]}=\frac{K_{a,R^{\prime }{{\mathrm{NH}}_{3}}^{+}}}{K_{a,\mathrm{RCOOH}}} \end{array}$$

(2)$$K_{am}=\frac{\left[{\mathrm{RCONHR}}^{\prime }][H_{2}O\right]}{\left[\mathrm{RCOOH}][R^{\prime }{\mathrm{NH}}_{2}\right]}$$

(3)$$\begin{array}{cc} K_{cond} & =\frac{\left[{\mathrm{RCONHR}}^{\prime }][H_{2}O\right]}{\left[{\mathrm{RCOO}}^{-}\right]\left[R^{\prime }{{\mathrm{NH}}_{3}}^{+}\right]}=K_{exch}K_{am} \end{array}$$

(4)$$\begin{array}{cc} K_{cond} & =\frac{K_{a,R^{\prime }{{\mathrm{NH}}_{3}}^{+}}}{K_{a,\mathrm{RCOOH}}}K_{am} \end{array}$$

The overall free energy change is positive (a typical value of $\Delta G_{cond}^{\circ }$ is ca. +20 kJ/mol or $K_{cond}$ ca. 3 × 10^−4^). It is the charged nature of the reactants in water that is the thermodynamic reason for making the free amino acids condensation an endergonic reaction. Work should be done to remove the charge from the carboxylate and ammonium ions in water (this depends on the p$K_{a}$ values of the reacting groups, but typically, $\Delta G_{exch}^{\circ }$ ca. +30 kJ/mol, $K_{exch}$ ca. 4 × 10^−6^). It results that the pure aminolysis reaction (RCOOH + R′NH_2_) is per se not unfavored ($\Delta G_{am}^{\circ }$ ca. −10 kJ/mol; $K_{am}$ ca. 60, as in the case of ester aminolysis; see below).

The direction of the condensation reaction can be forced to the right (toward the condensation product) using the ratio between the chemical species (thermodynamic control) [84]. In water-rich environments, hydrolysis prevails, and in water-poor environments, condensation becomes more favorable. Using a large molar excess of the reactants, or removal of one of the products by precipitation, evaporation, specific complexation, phase separation, or migration to another phase are the tactics for carrying out the condensation reaction against the unfavorable $\Delta G_{cond}^{\circ }$.

A second possibility to circumvent the hostile thermodynamics of peptide bond formation from free amino acids focuses on the use of ‘activated’ substrates (i.e., better acyl donors, such as carboxylic acid esters or similar compounds) (Figure 3b). Ribosomes catalyze the peptide bond formation in this way. The aminolysis reaction of an ester is thermodynamically favored (at pH 7, typical $\Delta G^{\circ }$’ ca. −10 kJ/mol, as deduced from thermodynamic values [85]), due to the greater bond strength and stability of amides when compared to esters, but catalysts are nevertheless important to overcome the kinetic barrier typically involved in the breakdown of the tetrahedral intermediate. This second strategy (kinetic control) [84] has been exploited in the application of enzymes in organic synthesis [86,87].

With these notions in mind and inspired by the principle of microscopic reversibility, we began investigations into the possibility of using Ser-His as a catalyst of peptide bond formation [18]. In particular, we have combined the above-mentioned strategies, i.e., activation of the substrate and removal of the product. We have demonstrated that Ser-His catalyzes peptide bond formation between a *C*-terminus of an activated amino acid (*N*-protected) and the free *N*-terminus of another amino acid (*C*-protected). The activating group on the *C*-terminus is a carboxylic acid methyl or ethyl ester (Figure 4). It should be noted that these reactants are intended as models of origin of life-relevant activated substrates. Interestingly, whereas Ser-His is capable of catalyzing the coupling reaction, His-Ser is not, and neither are serine or histidine alone or mixed together. The same phenomenon has been observed in the hydrolysis reactions [21]. It seems that reversing the order of amino acids changes the three-dimensional arrangements of side chains, as well as their geometrical relation (Figure 1). These structural changes translate into a mismatching arrangement of functional groups necessary to perform catalysis.

The most extensively studied reaction has been the condensation between two amino acid substrates (“1 + 1” combination, to give a dipeptide): *N*-acetylated phenylalanine ethyl ester and leucine amide (Figure 4), which lead to the water-insoluble dipeptide *N*-acetyl phenylalanyl-leucinamide [18]. The reaction has almost no uncatalyzed background and proceeds rapidly (within minutes) when catalyzed by α-chymotrypsin. The phenyl ring of the acyl donor ensures an easy detection by UV absorbance. Usually, HPLC analysis is employed to monitor the reaction. In the presence of Ser-His, the reaction proceeds better at neutral-slightly alkaline pH (7–9); higher pH values favor the substrate hydrolysis. The latter is reduced by working at low temperature (5–15 °C), whereas it prevails at higher temperatures. Under optimized conditions, yields go from 20%–50%, mainly depending on the duration of reaction (up to 30 days).

Higher peptides were formed analogously. In addition to the “1 + 1” combination, other peptides were synthesized according to the “1 + 2”, “2 + 1” and “2 + 2” combinations, to give, respectively, tri- and tetra-peptide (Figure 5). These results illustrate the applicability of Ser-His as a potential peptide ligase of peptide fragments, to form longer peptides. What is more, the reaction was extended to peptide nucleic acid (PNA) monomer substrates, producing, reaction, PNA-dimer, trimer and tetramer in a single pot [18] (Figure 5, last line). In contrast to the other cases, the latter reaction proceeds by joining ‘bifunctional’ substrates, namely having an electrophile and a nucleophile moiety in the same molecule, so that the oligomerization is possible. Ser-His, therefore, can also promote elongation reactions by repetitive addition of monomers to a growing chain through the formation of amide bonds, which in principle should be easily extendible towards growing peptide chains.

Prebiotically-plausible conditions are usually understood as water environments in mild temperature regimes [88]. In such conditions, hydrolysis of the activated substrate is favored. This was indeed also observed under Ser-His catalysis. Thus, the two reactions (peptide bond formation and hydrolysis) are in competition with each other, as happens in kinetically-controlled peptide synthesis [84]. Usually, a large part of the acyl substrate (up to 50%) ends up as a free acid compound. The available data do not help for the determination of the mechanisms underlying coupling and hydrolysis reactions, but the presence of a covalent complex between Ser-His and the acyl substrate is a possibility under discussion. A small amount of a compound with unknown structure and a molecular mass corresponding to covalent adducts have been isolated [89] when H-Phe-OEt or H-Trp-OEt was incubated with Ser-His attempting to synthesize Trp or Phe oligomers (conditions of Figure 5, last line).

Under the explored experimental conditions, for the transfer reaction to occur, we have to ensure that the product of condensation has a low solubility in water (Figure 4 and Figure 5). This characteristic ensures that the forming product will drop out from the solution and form an insoluble precipitate, thus driving the equilibrium of the reaction towards the formation of a peptide bond even in an aqueous environment (Figure 4). Therefore, the established method allows for the synthesis of hydrophobic peptides, with the general rule being that the product needs to be more hydrophobic than either of the substrates. The bigger the difference in the partition-coefficient between the substrates and the product, the more efficient the synthesis. (Nota bene: One of the consequences of this fact is that although we could have a case of peptide catalyzed peptide synthesis (Ser-His that would catalyze its own formation), there is no straightforward way for achieving an autocatalytic circuit where the catalyst and the product are the same molecular species. The catalyst itself, Ser-His, is an extremely polar molecule with four hydrogen donating/accepting functional groups.)

Removing the product from the reaction mixture does not necessarily mean precipitation. It can also mean the product migration to a different phase. If the reaction Ac-Phe-OEt + H-Leu-NH_2_ is carried out in a two-phase system, for example in the presence of lipid vesicles (intended as protocell models), the interior of their bilayers will be the hydrophobic environment to which water-insoluble dipeptide can migrate [90]. The accumulation of the dipeptide in the membrane has been shown to change the membrane stability. This can lead to a competitive growth mechanism of liposomes containing the catalyst Ser-His (and thus, the produced dipeptide) over those that do not possess it. This situation has been heralded as an example of selective processes that could operate even before the creation of genetic material in prebiotic vesicles [90]. It is an example of something that has been dubbed ‘supramolecular selection’, a process that operates in the prebiotic ecology stage of prebiotic evolution and ensures the selection of certain macromolecular assemblies over others [91]. Finally, it should be mentioned that in another study, Ser-His was employed to promote peptide bond formation between two groups linked to the ends of a synthetic PNA stem loop structure [92], but in this case, its effectiveness as an organocatalyst was poor.

### 4.3. Phosphodiester Bond Formation

Encouraged by the promising results of Ser-His-catalyzed peptide synthesis, we asked whether similar reverse hydrolysis could be obtained with nucleic acids. Our campaign explored several different paths. We achieved success with an approach involving polymerization of imidazole-activated mononucleotides in a water/ice eutectic mixture where water is cooled below its freezing point, but above the eutectic point [19]. Under such conditions, most of the water is trapped in the form of ice crystals, and the rest of the solutes are up-concentrated in the remaining liquid microchannels. This strategy achieves two functions: first, it radically lowers water activity, as most of it is in the solid state and does not participate in the reaction; second, previously diluted reactants reach concentrations in which their interactions are facilitated. Another advantage is that at such low temperatures, (ca. −18 °C), the fragile imidazole-activated nucleotides have a prolonged half-life [93]. The imidazole activation method is widely used as a model for non-enzymatic oligomerization [88,94].

Although the achieved length of RNA oligomers has been limited to only a few nucleotides, the catalytic formation of the phosphodiester bond by Ser-His has been well studied, and the proposed mechanism of the catalysis has been reported thanks to the isolation of an intermediate complex [19,20] (Figure 6).

The reaction proceeds much like other RNA polymerizations from imidazole-activated nucleotides. The presence of an imidazole ring covalently attached to 5′ phosphorus draws electron clouds away, thus allowing for a partial positive charge to form on the atom. It is subsequently attacked by 2′- or 3′- nucleophilic oxygen from 2′- or 3′-hydroxyl of another nucleotide. Imidazole is released, and a phosphodiester bond is formed. The mechanism is very similar in the presence of Ser-His. The initial step is a transamination reaction with imidazole being replaced by the Ser-His histidine ring. Because of structural similarity, the Ser-His-capped nucleotide is also an activated form of the nucleotide, like 2-methyl-imidazole, or 2-amino-benzimidazole, or 1-methyl-adenine [94]. The Ser-His-capped intermediate reacts with all nucleophiles present in the system to give the desired dinucleotide, a pyrophosphate dimer, and the unactivated nucleotide, respectively (Figure 6). It can also react with the free terminal amino group of Ser-His to give a *N*-phosphate peptide-nucleotide adduct. It is unclear whether the reaction of the Ser-His-capped nucleotide with the various nucleophiles proceeds by intramolecular catalysis (e.g., because of specific structural arrangement allowing bound Ser-His to actively take part in the reaction), intermolecular catalysis (e.g., by the involvement of a second Ser-His molecule) or just because the presence of the Ser-His cap as the leaving group is more effective towards polymerization than the bare imidazole.

The involvement of all other Ser-His functional groups (Figure 1) cannot be excluded. Indeed, Ser-His derivatives, such as *N*-acetyl Ser-His and Ser-His amide, as well as the amino acids Ser and His, as well as their mixture, do not display any catalysis. This implies that the complete dipeptide structure is important, not just a specific functional group(s). Note, however, that Ala-Ser also catalyzes phosphodiester bond formation, although with much less efficiency when compared with Ser-His. This demonstrates the importance of the serine hydroxy group.

### 4.4. A Tentative Discussion on the Catalytic Mechanism of Ser-His

Not much is known about the detailed mechanisms of the two synthetic reactions that are catalyzed by Ser-His (the formation of peptide bonds and phosphodiester bonds). The mechanism emerging from Ser-His catalyzed phosphodiester bond formation in Figure 6 is quite different from that which would be expected (cf. Figure 2). Here, an intermediate has been isolated with the imidazole ring, rather than the serine hydroxy group, involved in a covalent bond with the acyl donor. It is again a case of nucleophilic catalysis, but based on Ser-His imidazole, not on the Ser-His hydroxyl group. The essential role of the nucleophilic (unprotonated) imidazole in this mechanism is supported by the fact that the reaction does not occur below pH 6 and that when the Ser-His imidazoyl *N^ϵ^* is methylated, the resulting compound does not function as a catalyst.

Much less is known about the mechanism of peptide formation (Section 4.2). In this case, the pH dependence of peptide formation has been determined [18] for the system of Figure 4. It follows a neat sigmoidal profile with an inflection point at pH 7.85 (Figure 7), suggesting that the reaction product is generated by a mechanism where the deprotonation of an acid moiety with a p*K_a_* around such a value is fundamental. Although it is not surprising that in many proteins the imidazole p*K_a_* can differ (also largely) from the standard value of about six, depending on the micro-environment [95], in this specific case, it seems that the protonation status of imidazole of Ser-His does not play a key role. The three Ser-His p*K_a_* values are reported to be 2.42 (carboxyl), 6.71 (imidazole) and 7.70 (amino) [96]. It follows that if the rate-determining step is the initial interaction between Ac-Phe-OEt and Ser-His, the Ser-His terminal amino group might act under the general base catalysis scheme, facilitating the nucleophilic attack of another Ser-His molecule to the ester. If, on the other hand, the rate-determining step is the attack of the leucinamide terminal amino group to the carbonyl carbon, the dominant factor is the availability of such unprotonated amino group on the *N*-end of the H-Leu-NH_2_. More detailed studies are needed to clarify these mechanistic aspects. In any case, the mechanism of Ser-His-Asp action is expected to be the same as Ser-His with the carboxyl group of Asp and the carboxyl *C*-end of Ser-His performing a similar function in the overall mechanism.

### 4.5. Molecular Modeling Studies: What Can We Learn?

As we have mentioned, although Ser-His is an organocatalyst for hydrolysis and condensation reactions, neither the isomer His-Ser, nor the His + Ser mixture have a similar activity. To investigate the atomic bases of the different properties of His-Ser and Ser-His dipeptides, molecular dynamics (MD) simulations in explicit solvent were performed at 300 K on the respective molecular models, considering both tautomeric forms of monoprotonated histidine (protonated either on the *N^ϵ^* or on the *N^δ^* ring nitrogen). In detail, after energy minimization of the models, the temperature of the system was gradually increased from 0–300 K in 100 ps, and the systems were simulated for 1 ns at 300 K in the NVT ensemble. The final conformations obtained after 1 ns are shown in Figure 8.

As can be seen from the images, the four models display distinct conformations dictated by their specific stereochemistry. In detail, both the His-Ser and Ser-His dipeptides in the *N^δ^*–H state (Figure 8a,c) display a weak electrostatic interaction between the *N*-terminal ammonium group and the *C*-terminal carboxylate, which is formed very early in the MD simulations. Such an interaction is never observed during the course of the simulations in the *N^ϵ^*–H dipeptide. In the case of *N^ϵ^*–H His-Ser dipeptide (Figure 8b), a strong electrostatic interaction between the dipeptide *N*-terminal ammonium group and the Ser hydroxyl group is formed soon after the beginning of the MD simulations and persists until the end of the simulation time. In principle, in this latter case, the observed conformation could lead, after deprotonation of the terminal −NH_3_^+^, to polarization of the serine O–H bond, which is required to generate the strong nucleophile needed for a serine protease-like catalytic activity. Indeed, this is reminiscent of the Lys-Ser catalytic dyad observed for instance in bacterial Type I signal peptidases [97]). Despite this favorable conformation, experiments show that His-Ser is inactive (pH 10). On the other hand, *N^δ^*–H Ser-His (Figure 8c) adopts a folded conformation where the Ser hydroxyl group and the His imidazole ring approximately faced each other, but no specific intramolecular interactions can be recognized from the model. Note that previous work [81] has shown that *N^δ^*–H Ser-His has a conformation similar to that which has been found by us. Interestingly, no evidence for a short and directional intramolecular O−H····N hydrogen bond can be obtained from these 3D models. The conformation of *N^ϵ^*–H Ser-His (Figure 8d) resulted in being almost extended.

Therefore, MD simulation results do not provide a straightforward explanation for the occurrence of a serine protease-like catalytic activity in the Ser-His dipeptide as opposed to the His-Ser dipeptide. On the basis of this result, we can put forward the hypothesis that the active species responsible for the Ser-His catalytic properties might not be a monomeric, but rather an oligomeric, most likely dimeric, species in which a proper catalytic triad may be formed.

## 5. A Peptide “Fragment Condensation” Scenario

In Section 3, we have seen some of the most common pathways for the origin of peptides in prebiotic conditions. Summarizing, these are: condensation at high temperature or by water removal cycles, salt-induced condensation and oligomerization of activated monomers. According to this view, a pool of amino acids and oligopeptides of various lengths (typically <10-mers) and diverse sequences might be considered as the starting point for the emergence of interactive reaction networks, peptide-membrane and peptide-nucleic acid interactions and the usage of shorter peptides as feeding material for the synthesis of longer peptides. Organocatalysis helps here.

Some of these spontaneously-formed peptides could be endowed with special structural features, like with Ser-His, and could be capable of catalyzing the condensation of the other peptides, according to a fragment condensation pattern, which is reminiscent of the practice of peptide ‘chemical ligation’ operated by proteases [98,99]. Fragment condensation foresees the combination of small peptide oligomers to rapidly converge into large chains. Chain elongation, the biochemical way to synthesize biopolymers, is instead based on the growth of a chain by sequential addition, one after the other, of building blocks to one of the two chain extremes. To produce an 80-mer, 79 chain elongation steps are needed; starting from 10-mers, instead, seven fragment condensation steps are required.

The potential role of fragment condensation in generating long biopolymers has not been widely discussed in the literature, probably because most of the studies focus on mimicking the biological synthesis of biopolymers, which proceeds via template-directed chain elongation. Nucleic acids and protein elongation mechanisms rely on the same recognition strategy, based on how complementary the nucleotides are. On the other hand, one could speculate that primitive peptide fragment condensation is totally random or that it is guided by molecular recognition (e.g., peptide-peptide or peptide-catalyst). The latter could act as a selection rule in the combinatorial space of potentially reacting peptides. If fragment condensation had played a role in prebiotic times, then that role would have been functional, endowing the early molecular systems with the first long-enough (and thus, likely folded) proteins . Such a scenario could have anticipated and facilitated the path toward the more sophisticated template-based chain elongation.

Fragment condensation of primitive peptides, unfortunately, lacks experimental studies, but it is per se worthy of more attention (note, however, that a protease-catalyzed oligomerization of a native protein segment in neat aqueous solution has been reported [100]). A qualitative account has been recently depicted by Luisi [101] and illustrated by means of an artificial procedure that led to one folded peptide (44 amino acids long) [102].

The scenario is based on combinatorial peptide ligation combined with a selection rule that favors only a small number of produced peptides to be processed further along in the next step. In detail, starting from a pool of spontaneously-formed peptides, the members of the pool react with each other in a combinatorial way by a fragment condensation mechanism. Such reactions can be catalyzed by a few catalysts, such as Ser-His or other similar compounds, which might themselves be components of the pool or have been generated after fragment condensation (examples are known of peptide catalysts produced by combinatorial chemistry [103,104]). For example, the full combinatorial coupling of ten 10-mers theoretically gives a hundred 20-mers. Next, the produced 20-mers react with 10-mers or with 20-mers to give 30-mers or 40-mers, etc. (Figure 9).

Overlaid on this growth, which quickly increases the peptide length and thus the propensity to fold into three-dimensional compact structures, one can imagine selection rules that reduce the population of peptides participating in growth, for example by precipitation, degradation, incapacity to bind to the catalysts or by having partially hidden terminal groups that do not allow for further elongation. Moreover, combinatorial coupling could not involve all possible reaction partners with a similar efficiency. For example, peptides that strongly interact with each other, due to multi-point molecular recognition, might be favored over the others. The net result is that given a set of environmental conditions (pH, temperature, salinity, presence of denaturing agents, presence of surfaces), the possible large number of potential products is reduced and plausibly converges to a few compounds. These will be long, folded peptides with potential catalytic activity, not only toward fragment condensation, but also with respect to other reactions.

This hypothetical scenario has not been proven experimentally, but operating by the Merrifield synthesis, it has been shown how it could work in practice. Chessari et al. [102] showed that eight random 10-mer peptides, whose random sequences were generated by computers, were divided into two groups and combined to give sixteen 20-mers, which were all synthesized individually and subjected to solubility test. The four more soluble ones were next combined with four other 20-mers to give sixteen 40-mers. Two of these 40-mers, which were partially soluble, were further elongated to the *N*-terminus with a four hydrophilic amino acid tail (DDEE), to give two 44-mers, one of which was soluble in water (Figure 10a). Circular dichroism indicates ca. 22 residues in α-helix conformation, in agreement with simple predictions based on primary sequence analysis. Three-dimensional modeling (ROKKY protein folding suite [105]) suggests that the peptide folds as three α-helices around a hydrophobic core (Figure 10b).

The fragment condensation scenario is based on reverse proteolysis (peptide ligation). We have seen that Ser-His, a dipeptide that can be used as a model for the primitive protease/peptide ligase compound, requires the activation of the *C*-terminus. Thus, as we have remarked above, activated molecular partners are necessary for a realistic fragment condensation scenario. Until now, efficient fragment condensation with the help of Ser-His was only obtained from protected substrates (Figure 5), thus precluding further condensation of fragments. Obviously, for condensation steps to proceed further, we need to ensure that the reaction schemes described in Figure 5 as “1 + 1 + ...”, which result in unprotected products, advance in a more efficient manner. The low solubility of the products is the other related Ser-His requirement, which is needed to shift the equilibrium toward the peptide. This contrasts with the fragment condensation scenario, as the precipitation of a large part of the peptide library is intended as a step to eliminate, rather than selecting, coupling products. However, as demonstrated by Adamala and Szostak [90], if a reaction is carried out in a two-phase system, for example in the presence of lipid vesicles, lipid bilayers are the hydrophobic environment to which water-insoluble product can migrate. This possibility actually enriches the fragment condensation scenario by including lipid vesicles (for instance, fatty acid vesicles that are considered the most plausible protocellular structures) and membrane-associated peptides: a potentially fruitful peptide/lipid cooperation.

In this context, it should be recalled that studies have shown that molecular crowding, which has been found in some vesicle systems [106,107,108], favors reverse proteolysis by volume exclusion effects [109]. The lipid membrane can favor growing reactions only inside the liposome lumen, owing to its semi-permeability, and provides a matrix for the peptide formation at the same time [110]. For example, it has been suggested that hydrophobic peptides could possibly concentrate on boundaries between lipid domains [111].

The experimental verification of these potentially prebiotic mechanisms is challenging, but complementary information could be achieved by in silico approaches, e.g., molecular modeling combined with stochastic simulations which could be helpful to explore populations of random peptides, their condensation and the outcome of selection rules.

## 6. Other Potential Functions of Small Peptides

In modern proteins, cationic peptides, rich in arginine (Arg) and lysine (Lys), are known to strongly interact with RNA and DNA [98,112]. Nuclear DNA binding histones and protamines are mostly composed of such cationic proteins [113,114].

Shorter oligopeptides rich in Arg and Lys residues are also capable of interactions with nucleic acids [98,115]. For example, Arg-Arg-Arg tripeptides can aid DNA condensation via electrostatic interactions [112]. Poly-Arg and poly-Lys residues form salt bridges with nucleic acid and provide charge shielding of the nucleic acid backbones [116].

Arginine-rich peptides are prebiotically plausible, with several routes of the synthesis of arginine under prebiotic conditions [117,118]. It is possible that the cationic peptides could have a stabilized fold and therefore aid the function of the nucleic acids employed in the early life metabolism.

The emergence of self-replicating nucleic acids likely required some form of templating of nucleic acid synthesis [119]. One obvious templating surface in a protocell would be a bilayer membrane. However, both RNA backbone and lipid headgroups are negatively charged, making direct templating interactions impossible. It has been recently shown that hydrophobic peptides can promote localization of RNA to the liposome membrane, therefore providing templating surfaces for RNA activity [120]. Small hydrophobic peptides themselves can also interact with the lipid bilayer membranes. As mentioned, the small hydrophobic peptide AcPheLeuNH_2_ can localize into the fatty acid bilayer membranes. This causes the vesicles containing the peptide to grow at the expense of vesicles without the peptide. Furthermore, AcPheLeuNH_2_ can be synthesized by the Ser-His dipeptide. Combined, this creates a system where encapsulated small peptide catalysts can impact the fitness of model protocells, providing insight into the emergence of Darwinian selection [90].

Amino acids, especially neutral and hydrophobic ones, can cross simple, prebiotically-plausible fatty acid membranes [121]. However, as early life moved from leaky and unstable fatty acid membranes to modern phospholipid membranes, the permeability decreased sharply [122]. In the absence of active membrane transport, this created a problem in substrate delivery for the synthesis of peptides inside the protocell. It has been demonstrated that pH-gradients mediated amino acid transport and provided a solution to this problem [121]. This suggests that, in protocells, fatty acid membranes and encapsulated catalysts could have co-evolved after the earliest stages of the history of life [123].

By using a model system (Boc-tryptophan pyranine ester as the acyl donor and tryptophan benzylamide as the nucleophile), it was shown that the selective permeability of liposome membrane intrinsically facilitates the accumulation of condensation products inside liposomes [110]. Moreover, it has been suggested that boundaries between lipid domains might lead to peptide accumulation, leading to optimal conditions for further peptide reactions [111].

## 7. Looking for Short Random Primordial Peptides with Catalytic Functions

The role of amino acids as catalysts for some chemical reactions has been greatly emphasized in synthetic organic chemistry with the advent of organocatalysis, and recent studies point to short catalytic peptides as minimalistic forms of enzymes [28], revealing the potential of the still unexplored huge pool of short random peptides in organic synthesis and in prebiotic chemistry scenarios.

Ser-His, whose activity has been discussed in detail in this review, is just one example of these to-be-discovered peptides. It should be remarked that histidine is not generally considered a prebiotic amino acid. However, its synthesis in simulated prebiotic conditions has also been reported [43]. Although the detailed mechanism of Ser-His activity has not been clarified yet, this does not weaken the range and the relevance of the reported reactions. In particular, that a simple dipeptide can catalyze an important class of reactions, even if with low efficiency, is an important affirmation for prebiotic chemistry, one that suggests that further investigations in the field of peptide organocatalysis are warranted.

Peptides whose length is between 10 and 40 residues (and more) can be formed, at least in principle, by spontaneous polymerization of activated amino acids followed by fragment condensation, with the help of short peptide catalysts that are themselves components of the peptide pool. Not much is known about co-oligomerization of amino acids to form short peptides (e.g., up to the 10-mer) and about the nature of the resulting sequences (for example, see [124,125]). Are these really random? What did early peptides look like? Peptides, in contrast to nucleic acids, are stable and year-persistent molecules in the absence of catalysts (the half-life for uncatalyzed peptide hydrolysis has been estimated to be 450 years [126,127]). The gap between Miller-type experiments, referring to the accumulation of monomeric building blocks and even moderately complex scenarios with systems of molecules interacting in diverse ways, perhaps in the presence of membranes, is actually a relevant one. In this context, we are still missing not only experimental evidence, but also a conceptual framework.

On the other hand, the exploration of fully-random peptide libraries, looking for folding and catalysis, is challenging due to the vastness of the sequence space and to the peculiar physico-chemical behavior of peptides, which is hardly predictable. Attempts to investigate a library of 10^9^ random 50-mer peptides have been reported, resulting in a sub-population of peptides that are partially resistant to thrombin digestion, hence potentially folded [128,129]. Other reports also suggest that peptides formed by random sequences [31], or polar/non-polar alternate sequences [130], or from a reduced amino acid alphabet [131] can fold in well-defined three-dimensional structures; hence, they would be potentially catalytic. Furthermore, the possibility of peptides that self-assemble in the form of a complex that has a function (only as a complex) [132,133] should not be discarded. The results and the considerations presented here, therefore, actually call for further theoretical, experimental and in silico investigations on short peptides with random amino acid sequences, an enormous reservoir of molecular diversity. Finding new catalytic peptides would be of significance for understanding the origin of life, and it can also have a potentially high impact in synthetic chemistry and biotechnology. Prebiotic chemistry is the field where organocatalysis is such a native and powerful concept that it surely will be determinant for explaining many still unsolved questions (for example, see Smith and Morowitz contribution in [134]).

Addendum: While writing this review, we have discovered a very recent paper where the esterase/peptidase Ser-His activity has been re-examined and questioned based on additional tests [135]. In addition to *p*-nitrophenyl acetate, the authors have assayed the Ser-His hydrolytic function with fluorescein esters and rhodamine amides at pH 6, reporting a non-statistically-significant difference between control and Ser-His containing samples.

## Figures and Tables

**Figure 1 life-07-00019-f001:**
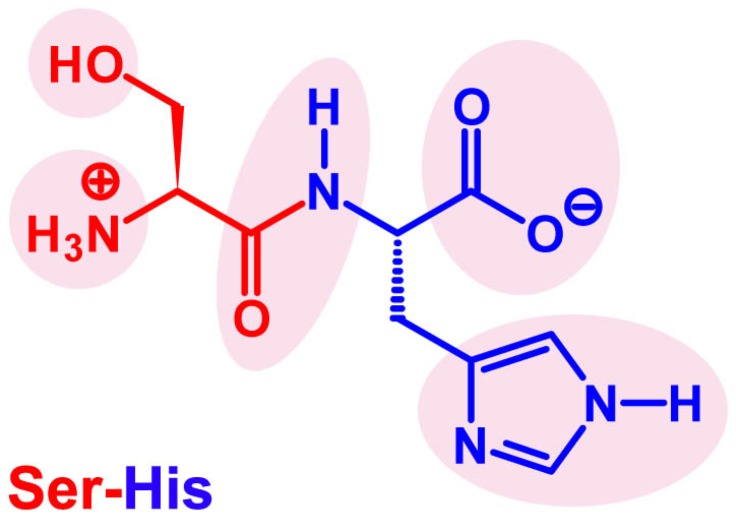
Chemical structure of Ser-His with evidence of its several functional groups. The imidazole ring is shown in its neutral *N^ϵ^*–H tautomeric form.

**Figure 2 life-07-00019-f002:**
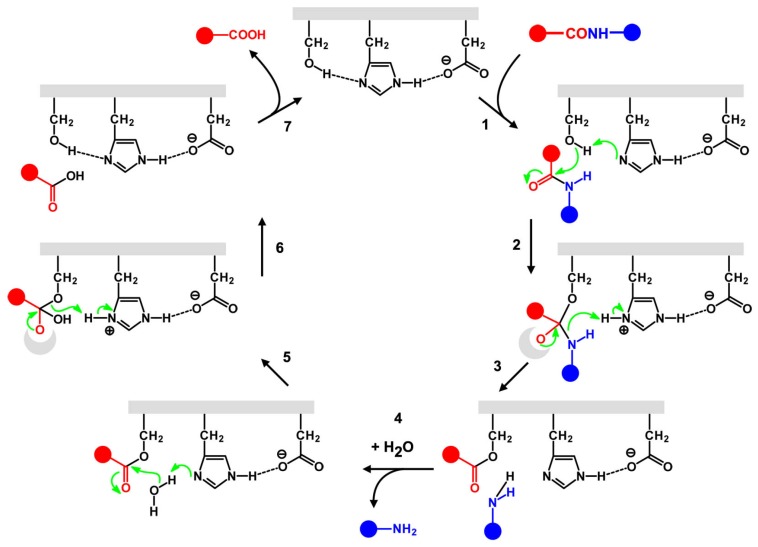
Mechanism of serine protease. The catalytic triad Ser/His/Asp acts in a concerted manner and cleaves the peptide bond in two steps. The acyl group is firstly transferred to Ser hydroxyl oxygen, then to water. See details in the text. The dashed lines indicate favorable interactions between the negatively-charged aspartate residue and the positively-charged histidine residue, which make the histidine residue a more powerful base. Note the ‘oxyanion hole’, which stabilizes via hydrogen bonding with amide N–H the negative oxygen of the tetrahedral intermediate [70].

**Figure 3 life-07-00019-f003:**
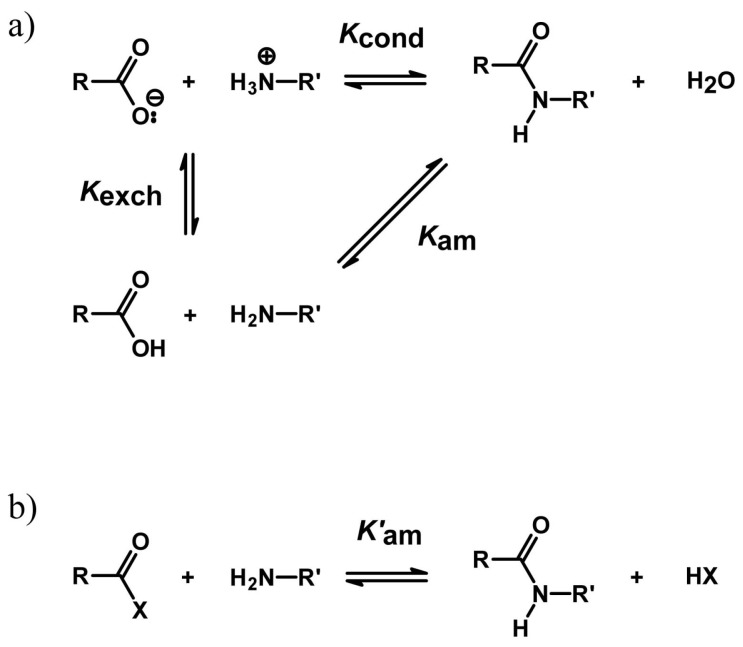
Formation of peptidic amide bond. (**a**) Condensation of free amino acids in water, where the reacting groups (terminal carboxylate and terminal ammonium) are charged and unreactive. A proton transfer from the ammonium group to carboxylate would lead to a reactive couple undergoing aminolysis. However, actually, $K_{exch}$ favors the charged compounds. As $K_{cond}$ = $K_{exch}$
$K_{am}$, the main thermodynamic barrier to amino acid condensation in water is their presence as a charged state, whereas $K_{am}$ would be in favor of the amide. (**b**) Aminolysis of the activated acyl derivative. In contrast to free amino acid condensation, ester aminolysis is thermodynamically favored (typically, $K_{am}^{\prime }$ 50–60), but the reaction pathway may be hindered by a kinetic barrier; and if carried out in water, hydrolysis competes with aminolysis.

**Figure 4 life-07-00019-f004:**
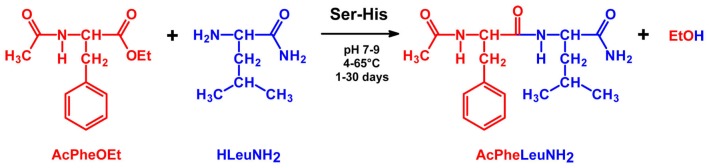
Ser-His catalyzed peptide formation [18]. *N*-protected and *C*-activated phenylalanine residue (AcPheOEt) reacts with *C*-protected leucine residue (HLeuNH_2_) to give ethanol and the desired dipeptide (AcPheLeuNH_2_) in a 20%–50% yield (depending on incubation time, up to 30 days), at pH 7–9 (Britton–Robinson buffer was used, including acetate, phosphate and borate). The reaction has been tested from 4–65 °C and runs well also in the presence of cupric ions and urea. Note that AcPheLeuNH_2_ precipitates in the reaction conditions, shifting the equilibrium to the right. As a comparison, if α-chymotrypsin is used instead of Ser-His, the reaction occurs in a few minutes. Together with aminolysis, AcPheOEt hydrolysis is observed (AcPheOH).

**Figure 5 life-07-00019-f005:**
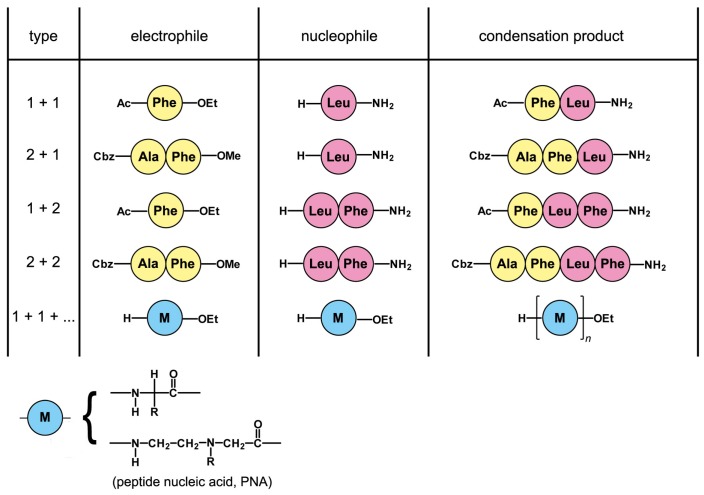
Schematic representation of the peptide-forming reaction catalyzed by Ser-His [18]. In addition to the “1 + 1” case, yielding a dipeptide, other combinations have been explored, up to the formation of tetrapeptide by a “2 + 2” condensation scheme. Moreover, by using “bifunctional” substrates, namely peptides having both an electrophile moiety and a nucleophile counterpart in the same molecules, the oligomerization of peptides was shown (H-Trp-OEt and H-Phe-OEt, up to the dimer, *n* = 2) and peptide nucleic acids (PNAs) units (up to a tetramer, *n* = 4).

**Figure 6 life-07-00019-f006:**
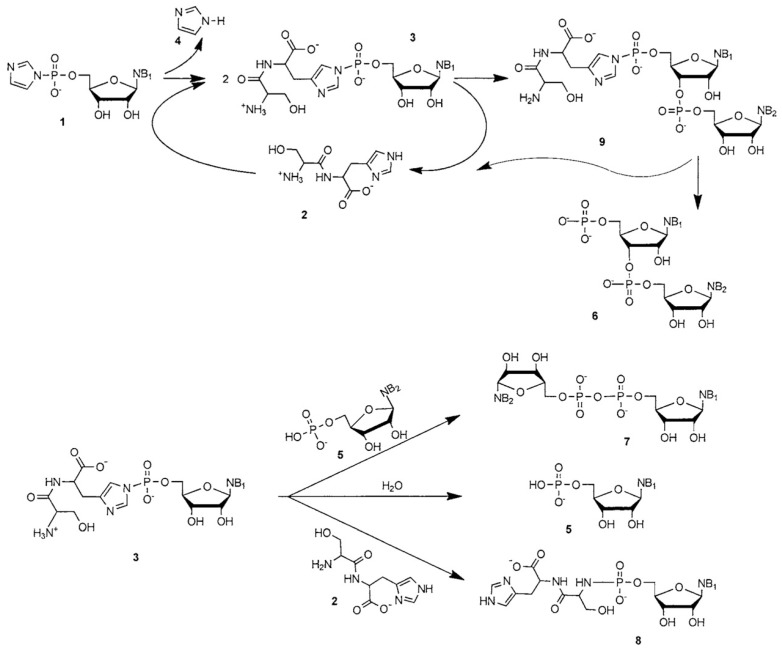
Proposed reaction scheme. Top: The oligomerization of imidazole-activated nucleotides (**1**) by Ser-His (**2**) involves the creation of a covalent linkage between dipeptide and mononucleotide (**3**), thereby releasing imidazole (**4**). Compound **3** rapidly forms and is then slowly consumed. It can be attacked by the 3′- or 2′-hydroxyl group of another nucleotide, thereby forming dimers **9** and **6** (only 3′–5′ bonds shown). Bottom: alternatively, it can be converted into side products: pyrophosphate dimer **7**, ribonucleotide **5** and an inactive, stable complex **8** are formed upon nucleophilic attack by a 5′-phosphate of **5**, water or the *N*-terminus of dipeptide **2**, respectively. Reproduced from Wieczorek et al. [19] with the permission of John Wiley and Sons.

**Figure 7 life-07-00019-f007:**
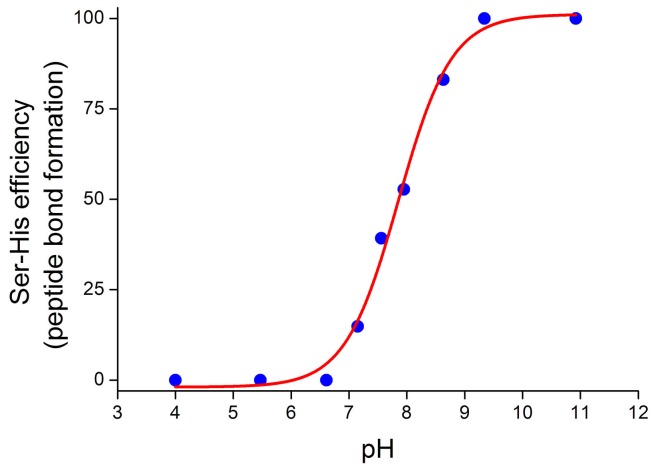
Profile of the Ser-His efficiency in promoting the aminolysis reaction shown in Figure 4. The inflection point of the sigmoidal curve, as obtained by non-linear fitting of experimental data, is at pH 7.85 [18].

**Figure 8 life-07-00019-f008:**
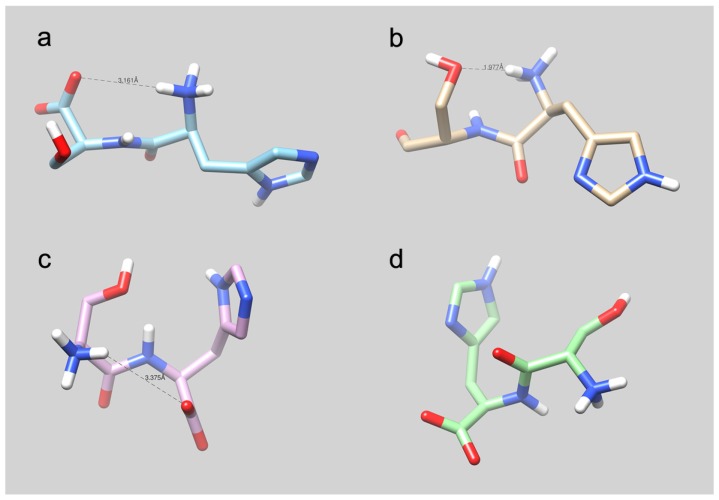
Molecular models of (**a**) *N^δ^*–H His-Ser, (**b**) *N^ϵ^*–H His-Ser, (**c**) *N^δ^*–H Ser-His and (**d**) *N^ϵ^*–H Ser-His, as obtained after 1-ns molecular dynamics simulations in explicit solvent.

**Figure 9 life-07-00019-f009:**
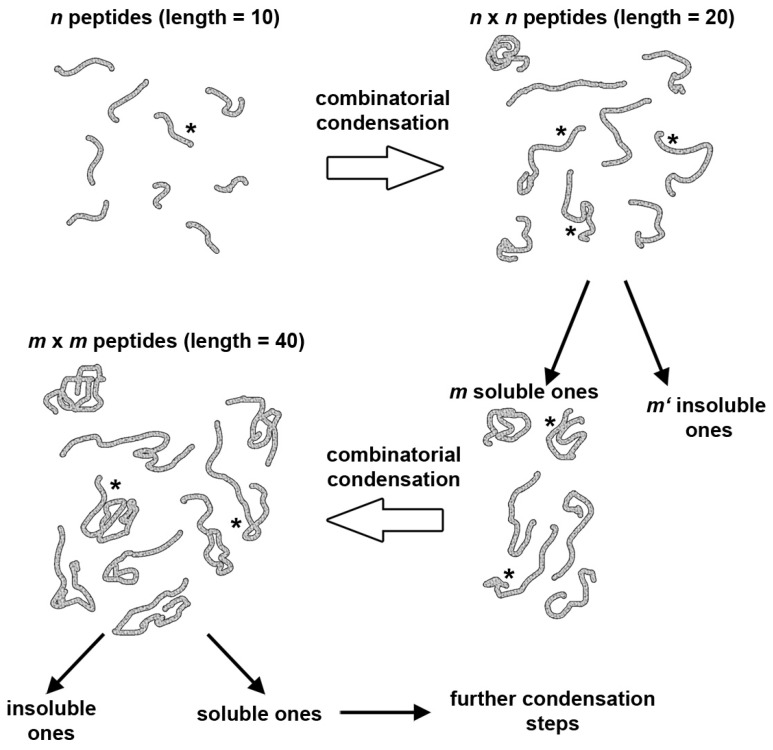
Peptide fragment condensation (adapted from [101]). Condensation scheme of *n* peptides (10 amino acids long) to yield ideally $n^{2}$ peptides (20 amino acids long), of which only *m* are soluble, and undergo further random fragment condensation yielding $m^{2}$ peptides (40 amino acids long), of which only a few will be water soluble. By continuing this procedure, we reach long chains that are soluble and that can be seen as the product of a prebiotic molecular evolution. The asterisk indicates a catalytic center, capable of inducing peptide synthesis. The synthesis can be in principle also induced by an external catalytic peptide. Reproduced from [20] with the permission of the authors.

**Figure 10 life-07-00019-f010:**
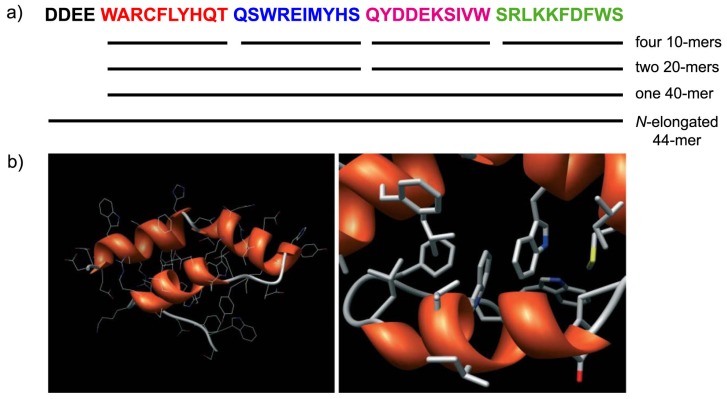
Novel folded peptide by fragment condensation [102]. (**a**) The sequence of a 44-mer peptide deriving from fragment condensation of four random 10-mers, whose sequence was randomly generated. Starting from the 10-mers, a library of 20-mers was constructed by Merrifield synthesis, and the soluble products were selected for the next condensation cycle. To the final 40-mer, which was partially soluble, a hydrophilic tail (DDEE) was added to the *N*-terminus, yielding a soluble 44-mer peptide, which was subjected to circular dichroism investigation. Spectroscopic data indicated that about 50% of the residues were in α-helix conformation. (**b**) Model of the three-dimensional structure of the 44-mer peptide (obtained by the ROKKY protein structure suite). Left: global view; right: detailed view of the hydrophobic core. Reproduced from Chessari et al. [102] with the permission of John Wiley and Sons.

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
