# Peer review of "Small and Random Peptides: An Unexplored Reservoir of Potentially Functional Primitive Organocatalysts. The Case of Seryl-Histidine"

_life, 2017, doi:10.3390/life7020019_

Round 1

Reviewer 1 Report

Version:1.0 StartHTML:0000000190 EndHTML:0000003920 StartFragment:0000002375 EndFragment:0000003884 SourceURL:file://localhost/Users/pizzar/Dropbox/2017/REVs_etc./Life/Lifes'17rev.doc

The submitted Review covers what its title promises, is comprehensive with pertinent reactions described in full, is properly referenced and discussed and this reviewer would like to see it published. However, the manuscript also makes a very difficult reading in its introductory pages and, if just for that section, is not publishable as written.

The English is poor and uses too many circuitous and perplexing choices of statements that could and should instead be plain. This reviewer has added sticky notes and suggestions to pages 1-4 and, as shown, there are entire paragraphs in those four pages that can be cut or at least reduced and rewritten. The authors should now take it from there.

As a general suggestion, authors should consider avoiding odd emphases and terms such as: intriguing, fascinating, remarkable, huge and the like to describe reactions or results. Also, unless required by particular emphasis, it is generally good and more clear to use subject, verb and predicate in that order. Since this is a review, all references should be explained, if briefly, instead some early ones appear with barely one line of explanation (e.g., 47-49) and authors should either cancel or better describe them.

Author Response

The submitted Review covers what its title promises, is comprehensive with pertinent reactions described in full, is properly referenced and discussed and this reviewer would like to see it published. However, the manuscript also makes a very difficult reading in its introductory pages and, if just for that section, is not publishable as written.

The English is poor and uses too many circuitous and perplexing choices of statements that could and should instead be plain. This reviewer has added sticky notes and suggestions to pages 1-4 and, as shown, there are entire paragraphs in those four pages that can be cut or at least reduced and rewritten. The authors should now take it from there.

We would like to really thank the Reviewer for the time used to directly work on the manuscript, suggesting in explicit way the best formulation of some sentences, and improve the quality of writing. Starting from page 5, we have continued the manuscript editing with the help of a native speaker who modified the structure of the sentences in the remaining pages, as suggested.

As a general suggestion, authors should consider avoiding odd emphases and terms such as: intriguing, fascinating, remarkable, huge and the like to describe reactions or results. Also, unless required by particular emphasis, it is generally good and more clear to use subject, verb and predicate in that order.

Emphatic terms have been deleted.

Since this is a review, all references should be explained, if briefly, instead some early ones appear with barely one line of explanation (e.g., 47-49) and authors should either cancel or better describe them.

With respect to References, we would like to keep all references as they were initially introduced. In our manuscript there are references which are accompained by condensed concepts (like the first 5 references, for example), other which are briefly explained (for instance, ref. 16), and others which are fully explained (e.g., ref. 18). Describing in detail all references would produce a text that is just an elencation of all work on a subject, which was not the goal of our review. Our intent is not to review all possible work done in this area, but specifically discussing a well defined question. To do so, we mainly referred to papers that are directly/closely related to our thesis, and with minor details, to other papers that are cited for enriching our discussion. The shortly-commented ones, like ref. 1-5, for instance, are included for the sake of completeness. We believe that all references, even if not commented in detail, give to the readers enough information to decide whether a certain paper is interesting for them or not.

In order to avoid this misunderstanding, we have written in the Introduction that this review is not intended as a comprehensive review that describes in detail all possible work on prebiotic chemistry and on peptide-bond formation. In particular we have added:

Since we previously focused our attention on the role of peptides in these prebiotic scenarios [18-20], here we aim to discuss the literature regarding the discovery that a simple dipeptide, Ser-His (Figure 1), is capable of catalysing the coupling reaction of amino acid derivatives or nucleotide derivatives. We will then present a general view on what could have been the roles of short and random peptides in the origins of life. In doing this, we will cite selected papers of the origins of life literature. It should be clear that this review is not intended to be comprehensive of the large amount of work done on this subject. Some references will be just shortly commented, whereas only the most relevant for our discussion will be presented in detail. In addition to Ser-His related finding, screening of peptide libraries, organocatalysis mechanisms, molecular recognition, and interaction with primitive membranes are possible fields of inquiry for further development in this research.

Reviewer 2 Report

The review by Wieczorek et al. comprises a broad view of the possible role of peptide catalysis in the origins of life, and an original perspective about how recent results on the catalytic activity of the Ser-His dimer could lead to a new research line within the area. The authors are actually pioneers of this research line, and the topic of the review is clearly of interest. The article is written in a clear English, and the references are adequate to a major extent. Some interesting aspects, to this reviewer opinion, is how they put organocatalysis in the context of prebiotic chemistry, the fact that they suggest new research lines and a whole model of a peptide world where peptides of different lengths would have functions that will assist nucleic acids and compartments at the onset of life. The manuscript is therefore suitable for publication in Life, although some issues need to be further clarified and reconsidered. I list them below:

1) There are a couple of mistakes regarding the use of “symbol” characters: page 2, line 60; and page 4, line 119.

2)  In section 3, I miss a number of important points:

2.1) His is not in the list of “prebiotic amino acids” that the authors give, but it is crucial in the catalysts they study. Can the authors propose, of course based on the literature, any possible pathway to His, or some possible source of it?

2.2) In their revision of prebiotic syntheses toward oligopeptides, there are important recent papers missing: Murillo-Sanchez et al., 2016, Chem. Sci. 7, 3406; Jaker et al., 2015, Angew. Chem. Int. Ed. 54, 14564. These two papers are actually important because they treat oligopeptides in interaction with lipid membranes and oligonucleotides, respectively.

2.3) In page 5, paragraph 3, the authors parallel experimental research lines toward oligopeptide libraries with Kaufmann´s ideas on autocatalytic chemical networks. This is of course reasonable, but a great limitation of their model is that in most of the peptide prebiotic syntheses described up to know, only dimers and trimers are obtained. This limits significantly their possible functions, and so such current limitation should be expressed more clearly somewhere in the review.

2.4) Something that is completely missing in the review relates to the work developed on peptide-based replication networks, by Ghadiri and Ashkenasy, among others, and which is fundamental if one wants to parallel the proposed research line with Kaufmann´s models. Some of the Ashkenasy´s systems are actually catalytic: see e.g., Chem. Eur. J. 2016, v. 22, 6687.

3) Another current limitation in the route to the peptide world suggested by the authors, which should be express more clearly, is that most of dimers, trimers and tetramers up to know are obtained from protected amino acid monomers. This raises doubts about the efficiency of the chemical couplings connecting oligopeptides that should lead to the proposed “fragment condensation scenario”. The authors should elaborate more on this, otherwise this scenario is for the moment not really credible.

4) In pages 8-9, the authors are not very clear about the systems they refer to in the last paragraph of page 8 (peptides, right?) and in the first paragraph of page 9 (peptide nucleic acids, right?). This misunderstanding occurs in other parts of the review (e.g., last paragraph of page 12), and should be better clarified, as they are very different systems, with the only similarity that they are constituted by amide bonds.

5) Finally, the catalytic triad Ser-His-Asp is mentioned. A pertinent question in this respect is whether it could have similar catalytic roles in prebiotic chemistry, considering its ubiquity in hydrolytic enzymes. Some further elaboration in this respect would be nice.

Author Response

We thank you for the lucid comments that stimulated us to improve the manuscript. The detailed answers, which address all your points, and a revised manuscript (with highlighted changes) can be found attached as pdf file.

With best regards

P. Stano

Reviewer 3 Report

Stano and coworkers have written a review on the Ser-His dipeptide and its potential role as a primitive hydrolase/condensation catalyst in the primordial Earth environment. The review has several contentious aspects including the following:

His is considered a “phase 2” amino acid – i.e. it is an amino acid that does not appear to be synthesized easily by prebiotic chemical processes, and therefore requires the establishment of biosynthetic pathways to be created. Thus, its relevance for transitional chemistry in the origin of life is questionable.

The authors describe a molecular dynamics study of the Ser-His dipeptide (as well as the apparently non-functional His-Ser dipeptide) in order to identify a conformation that provides a plausible explanation for nucleophile activation of the Ser. Such studies fail to identify a structural basis for catalysis. This is somewhat problematic since the model system (a dipeptide) is comparatively simple and should permit adequate MD sampling to identify low energy conformations. Thus, the failure of the MD study calls the catalytic activity of Ser-His into question.

 A recent report on the evaluation of catalytic properties for the Ser-His dipeptide by Gellman and coworkers refutes the claim of catalytic activity for the Ser-His dipeptide.

However, each of these issues is mentioned in the review by Stano; thus, while there is controversy regarding the subject, the authors review stands as a scholarly work that notes such objections. The review includes many appropriate references. Thus, the review is a scholarly work on a controversial subject.

Author Response

(The authors gave the same response as above.)

Round 2

Reviewer 2 Report

The changes and corrections made by the authors in this new version of the article are satisfactory to this reviewer. The paper can now be accepted for publication as it is.

Author Response

We thank the referee for his/her positive evaluations.